# CD4^+^ T-Cell Plasticity in Non-Infectious Retinal Inflammatory Disease

**DOI:** 10.3390/ijms22179584

**Published:** 2021-09-03

**Authors:** Yi-Hsing Chen, Sue Lightman, Virginia L. Calder

**Affiliations:** 1UCL Institute of Ophthalmology, University College London, London EC1V 9EL, UK; yi-hsing.chen.14@alumni.ucl.ac.uk (Y.-H.C.); s.lightman@ucl.ac.uk (S.L.); 2Department of Ophthalmology, Chang Gung Memorial Hospital, Taoyuan 33305, Taiwan; 3College of Medicine, Chang Gung University, Taoyuan 33302, Taiwan; 4NIHR Biomedical Research Centre, Moorfields Eye Hospital NHS Foundation Trust, London EC1V 2PD, UK

**Keywords:** Th17/Th1 (CD4^+^IFNγ^+^IL-17^+^) cells, Th17/Treg (CD4^+^IL-17^+^FoxP3^+^) cells, regulatory (Treg) T cells, Th1 cells, Th17 cells, experimental autoimmune uveitis, uveitis, CD4^+^ T cell plasticity

## Abstract

Non-infectious uveitis (NIU) is a potentially sight-threatening disease. Effector CD4^+^ T cells, especially interferon-γ-(IFNγ) producing Th1 cells and interleukin-17-(IL-17) producing Th17 cells, are the major immunopathogenic cells, as demonstrated by adoptive transfer of disease in a model of experimental autoimmune uveitis (EAU). CD4^+^FoxP3^+^CD25^+^ regulatory T cells (Tregs) were known to suppress function of effector CD4^+^ T cells and contribute to resolution of disease. It has been recently reported that some CD4^+^ T-cell subsets demonstrate shared phenotypes with another CD4^+^ T-cell subset, offering the potential for dual function. For example, Th17/Th1 (co-expressing IFNγ and IL-17) cells and Th17/Treg (co-expressing IL-17 and FoxP3) cells have been identified in NIU and EAU. In this review, we have investigated the evidence as to whether these ‘plastic CD4^+^ T cells’ are functionally active in uveitis. We conclude that Th17/Th1 cells are generated locally, are resistant to the immunosuppressive effects of steroids, and contribute to early development of EAU. Th17/Treg cells produce IL-17, not IL-10, and act similar to Th17 cells. These cells were considered pathogenic in uveitis. Future studies are needed to better clarify their function, and in the future, these cell subsets may in need to be taken into consideration for designing treatment strategies for disease.

## 1. Introduction

Non-infectious uveitis (NIU) is a sight-threatening inflammatory ocular disease particularly affecting people of working-age [1]. The prevalence of the disease is relatively rare, affecting 9 to 730 cases per 100,000 people [2]. As a result of inflammation, 20% of patients were reported to develop vision loss [1]. The etiology of NIU can be attributed to either autoimmune with systemic involvement, autoinflammatory, or idiopathic [3]. Despite the heterogeneous causes of NIU, current management comprises of management of the inflammation with corticosteroids; exactly how they are given depends on the site of the inflammation. Posterior segment involvement is sight-threatening and requires immunosuppression of acute inflammation with corticosteroids, and a stepladder approach of introduction of other immunosuppressant drugs, whilst newly discovered biologics serve predominately as third line agents for those with more severe vision-threatening NIU or who failed previous treatments [4]. However, without fully understanding the root cause of each clinical subtype of NIU, development of personalized medicine in the use of immunosuppression is limited.

The sequestration of the eye from the immune system protects vision from inflammatory insults. The common feature in all types of uveitis is the breakdown of the blood-aqueous and blood-retinal barriers (BRB) with vascular leakage observed by fluorescein and indocyanine green angiography. It is followed by leukocytes entering the eye, presenting as inflammatory cells observed clinically as anterior uveitis, vitritis, retinitis, choroiditis, and vasculitis. The clinical observations were supported by evidence of mainly CD4^+^ T cells, as opposed to CD8^+^ T cells, infiltrating the retinae, as observed by histology [5]. Naive T cells, when activated by the antigen presenting cells (APCs), differentiate into distinct cytokine-producing effector CD4^+^ T cells, with IFNγ-producing Th1 and interleukin 17 (IL-17)-producing Th17 cells being dominant in NIU. These cells are also characterized by their transcription factor expression, with Tbet involved in Th1 cell development, and RORγt in Th17 cell development. These cells then recirculate back to the eye, induce Th1- and Th17-related responses, and activate myeloid cell responses, thus contributing to structural damage in the retinal tissues in NIU and in the experimental model (EAU) [6,7]. Further studies have shown that Th17 and Th1 cells may play a role during different stages of inflammation and in different uveitis entities [8].

Recent findings have suggested that certain mature CD4^+^ T cells are able to de-differentiate or trans-differentiate into other T cell subsets with multiple phenotypic characteristics in response to injury and inflammation [9]. This can occur between T helper (Th) and T follicular helper (fh) cells, for example, between Th1/Th2, Th2/Th17, Th17/Th1, Treg/Th17, Tfh/Th17, Th2/Tfh, and Th2/Th9 cells [10]. Plasticity between distinct CD4^+^ T cell subsets was reported in both human and mouse T cells, under certain conditions, in vitro [11] and in mice in vivo [12,13]. It was proposed that a diverse repertoire of antigen-specific CD4^+^ T lymphocytes is essential for the host to respond to emerging microbial threats, to allow secondary responses to previously encountered pathogens, to suppress immune responses after microbial clearance to avoid tissue damage resulting from inflammation, and may be the solution for evolutionary preservation [10,14,15].

It has been reported that CD4^+^ T cells, in particular Th1 and Th17 cells specific for retinal antigens, were both able to induce EAU in naïve mice by adoptive transfer [16]. However, little is known about the involvement of so-called plastic CD4^+^ T cell subsets in uveitis. The present study aimed to review the function and roles of plastic CD4^+^ T cell subsets in uveitis patients. We will focus on two of the subsets—Th1/Th17 and Th17/Treg cells—in uveitis, as Th2, Th9, and Tfh subsets have only a limited involvement in the pathogenesis of uveitis.

## 2. Evidence of CD4^+^ T Cell Subsets in Monophasic and Relapsing EAU Models

EAU is a CD4^+^ T lymphocyte-mediated autoimmune disease model that recapitulates many features of NIU and can be used to study the pathogenesis, specifically the cellular interactions to retinal-specific antigens from immunologically privileged sites [17]. Historically, EAU has been successfully induced with retinal extracts, purified protein, and synthetic peptides in rabbit, guinea pig, rat, and mouse models [17,18,19,20]. Our review focusses on induced rat and mouse EAU due to the details of T cell subsets that have been published (as given below). Since the specific T-cell response, in adoptive transfer models, is dependent on the phenotypes of the antigen-specific CD4^+^ T-cells injected, and the response in the passively induced EAU in genetically-modified mice depends on the genes modified, so data are only included from some of these models.

Only certain rat and mouse models of uveitis allow investigation of the immune mechanisms underlying relapsing and monophasic autoimmune reactions at the level of autoantigen specificity. The most commonly used antigens for the induction of EAU in Lewis rats include peptide R14 (aa 1169–1191), R16 (aa 1177–1191, ADGSSWEGVGVVPDV), and peptide PDSAg (aa 341–354), extracted from retinal S-Ag, which are emulsified in complete Freund’s adjuvant (CFA) and fortified with Mycobacterium tuberculosis (mTB) strain H37RA. These models induce an acute, clinically severe posterior uveitis with high incidence (up to 100%) [21,22]. PDSAg and R14 induce a pan-uveitis involving cellular infiltrations in the anterior chamber, as well as in the retina and vitreous, resulting retinal structural damage. R16 also induces anterior and posterior uveitis, although in a monophasic presentation [23].

Classical EAU in mice is mainly induced in B10.RIII and C57/Bl6 strains, using IRBP_161–180_ (SGIPYIISYLHPGNTILHVD) and IRBP_1–20_ (GPTHLFQPSLVLDMAKVLLD), respectively. The most susceptible mouse strain for EAU is B10.RIII, whilst C57/Bl6 mice are only moderately susceptible to EAU. These two mouse strains develop similar signs of posterior uveitis, including vasculitis, optic neuritis, retinitis, vitritis, and choroiditis. However, the time course of disease is very different between the two strains. The disease in B10RIII peaks at 14 days post induction (dpi) and resolves after 21 dpi, with B10.RIII mice presenting a more severe form of disease [20]. In contrast, in C57/Bl6 mice, the onset is later, peaking at 21 dpi, and retinal inflammation lasts several months. Recently, it has been reported that EAU in C57/Bl6 involves Th1-cell involvement in disease induction phase and a Th17-cell response at peak and later phases [8]. In addition, by administering a lower dose of IRBP in C57/Bl6 and B10.RIII mice, one quarter exhibited chronic uveitis lasting 6–8 months post induction and low-grade disease (<2) throughout [20]. These chronic models are characterized by prominent choroiditis, outer retinal structural damage, and less vitreous or retinal infiltrates [24]. IRBP-pulsed matured-DC injection induced posterior uveitis in B10.RIII mice with more pronounced retinal vasculitis, and less focal retinitis representing a less severe (autoinflammatory) disease than classical EAU with a prominent granulocytic rather than lymphocytic infiltration in the eye [25].

It is known that effector CD4^+^ T cell subsets express a functional T cell receptor (TCR) and develop tolerance for self-antigens within the thymus by a positive and a negative selection process before being released into the periphery and secondary lymphoid organs [10]. A newly developed TCR transgenic (Tg) mouse line (R161H), expressing a TCR specific for IRBP peptide on the B10.RIII background, generates 100% spontaneous uveitis by 2 months of age. The pattern of uveitis mainly involves vitritis initially before progressing to posterior uveitis, with minimal anterior segment inflammation developing [19,26,27,28]. Another model with which to investigate the role of central tolerance to retinal antigens in the development of spontaneous uveoretinitis is the Autoimmune Regulator knockout (AIRE-/-) EAU model. Many retinal antigens, including IRBP, are controlled by the negative selection of AIRE transcription factor for their expression in the thymus, which is supported by the discovery that the susceptibility of mouse strains to EAU correlates reciprocally with the amount of IRBP expressed in the thymus [29]. AIRE-/- mice develop a spontaneous and chronic-progressive multi-focal chorioretinitis at 5–6 weeks of age and scar formation at 10–14 weeks [30]. The ocular inflammation observed is, however, less aggressive than that seen in R161H mice [26]. A table summarizing the roles of different T cell subsets in each EAU model are detailed in Table 1.

## 3. Role of Th17/Th1 Cells in EAU

It has been reported that Th17 cells have substantial plasticity and readily acquire ability to produce IFNγ in addition to IL-17 [15,38]. Our previous data suggested that the ability of Th17/Th1 cells to produce both IL-17 and IFNγ correlates with RORγt^+^ and Tbet^+^ transcription factor co-expression [36]. This subset, so-called Th17/Th1 cells, is thought to be due to Th17 cell plasticity, facilitated through IL-12 or IL-23 stimulation [39,40]. In the context of infection, it has been reported that IL-1β serves as a pro-inflammatory regulator of Th17 cells for its re-direction to Th17/Th1 cells [41]. Human Th17/Th1 (IL-17^+^IFNγ^+^) cells are characterized by their expression of CD161, CCR6, IL-17 receptor E, *RORC*, and IL-4-induced gene 1 [38,42,43]. These Th17/Th1 cells are characterized by their resistance to corticosteroids in IRBP-induced EAU at peak disease, which may be due to a stable expression of multi-drug transporter type 1 (MDR1), an ATP-dependent membrane efflux pump [36]. It is supposed that Th17/Th1 cells lose their ability to produce IL-17 and become non-classical Th1 cells (expressing both RORγt and Tbet, but producing only IFNγ) following an IL-17/IFNγ double-producing phase [44]. In addition, these CD161^–^CCR6^+^ CXCR4^+^ non-classical Th1 cells could further progress to a fully differentiated Th1 phenotype under the control of transcription factor Eomes [44,45,46,47]. Nevertheless, a reverse plasticity from the presumed static Th1 to Th17 cell in response to TGFβ and IL-6 has been reported [47,48]. 

Trans-differentiation of CD4^+^ T cells into Th17/Th1 cells may be vital in protective and pathogenic immune responses [49]. It has been reported that Th17/Th1 cells are involved in the pathogenesis of Th17-mediated diseases, for example, in juvenile idiopathic arthritis [50], graft versus host disease (GVHD) [51], in skin lesions in Behçet’s disease [52], experimental autoimmune encephalitis (EAE) [53], Crohn’s disease [54], and in dry eye disease (DED) [55,56]. These Th17/Th1 cells were reported to be associated with β-cell autoantibody activity in human and animal models of type I diabetes mellitus and could be used as a biomarker [57]. In a DED mouse model, the existence of Th17/Th1 cells were found originated from Th17 cells and contributed to dry eye severity in addition to Th17 cells [55]. In a Th17-induced colitis mouse model of inflammatory bowel disease (IBD), Th17/Th1 cells were confirmed to participate in initiation of intestinal inflammation [58]. It has been reported that plasticity of Th17 cells may develop due to the absence of RORγt in the participation of transcription factor activation, via stabilizing positive feedback, thus relinquishing its original function, developing plasticity with the influence of environmental signals [59].

In previous studies investigating the natural history of CD4^+^ effector T cells using rat and mouse EAU models, it was reported that Th17 and Th1 cells are involved at different stages of inflammation (Table 1). The level of retinal Th1 and Th17 cells were comparable at uveitis onset and the levels remained similar throughout in a monophasic rat EAU model. Th17/Th1 cells were initially observed at a low level but increased two-fold at disease resolution [22]. However, the composition of CD4^+^ T cell subsets was divergent in a chronic relapsing rat EAU model. Th17 cells dominated during the initial phase of disease and decreased thereafter. Conversely, Th1 cells participated minimally at disease onset but became prevalent during resolution and relapse phases. Th17/Th1 cells remained at similar detection levels throughout [22]. In the monophasic and relapsing mice EAU models, Th1 cells were reported to participate in disease induction, whereas Th17 cells were increased during peak disease. There were about 10% of Th17/Th1 cells detectable at the peak of retinal inflammation [6,34,36,37]. In the chronic EAU mouse models, during the initial peak stage of inflammation, Th17 cells are more readily detected, whilst Th1 cells have limited function and only relatively few Th17/Th1 cells were observed [24]. In the DC-induced EAU model, more Th1 than Th17 cells were detected at peak disease during retinal inflammation, although Th17/Th1 cells were not investigated in that study [25]. It has been reported in an IRBP-specific TCR-transgenic EAU model (R161H) that only minimal levels of Th17/Th1 cells (<1%) were detected as compared to EAU in a wild type B10.RIII model (10–15%) at peak disease [28].

Using the R161H model, Th17 cells were found to become less pathogenic following exposure to IL-24, via a negative feedback mechanism through the IL-17A receptor and the NF-kB signaling pathway. However, IL-10-producing cells were unaffected by IL-24 [60], suggesting that Th17/Th1 cells could be regulated by IL-24 produced by Th17 cells. There is supporting evidence that these Th17/Th1 cells were generated by local inflammatory signals, rather than being induced in the thymus. In rheumatoid arthritis, human Th17/Th1 cells proliferate to regulate disease activity in response to TCR signaling, cytokine polyfunctionality and resistance to Treg suppression [11]. In both the AIRE-/- and R161H spontaneous EAU models in which central TCR signaling to retinal antigens is manipulated, minimal Th17/Th1 cells were indeed observed. In contrast, in experimental autoimmune encephalomyelitis (EAE), an autoimmune CD4^+^ T cell-mediated central nervous system inflammatory model similar to EAU, it has been hypothesized that PTX and CFA immunization induce early expansion and differentiation of encephalitogenic CD4^+^ T cells and promote the expansion of Th17/Th1 cells but not Th17/Treg cells [61]. Even if the cells were generated centrally, it has been noted that Th17/Th1 cells were able to migrate, using an in vitro CNS barrier endothelial cell model, at comparable levels to Th1 and Th17 cells [62], suggesting that the genetically manipulated EAU models are not appropriate for studying CD4^+^ T-cell plasticity.

## 4. Evidence of Th17/Th1 Cells in Uveitis

It has been reported that different clinical presentations of Behçet’s disease (BD) involve different T cell subsets. For example, Th17 cells have been shown to play a more dominant role in BD patients with associated NIU (ocular BD) and folliculitis [63]. In BD patients with oral ulcer involvement, it has been shown that the levels of Th17/Th1 cells, producing both IL-17A and IFNγ, were significantly increased in BD patients compared with healthy subjects [64]. In sarcoidosis with lung involvement, however, the presence of tissue CD4^+^Tbet^+^RORγt^+^CXCR3^+^CCR6^+^ T cells was associated with a better prognosis [65]. Another study investigating peripheral blood from active or quiescent NIU patients all receiving systemic corticosteroids, compared to healthy controls, reported that levels of CD4^+^Tbet^+^RORγt^+^ T cells were not associated with disease status [66]. These studies, however, were limited since the cytokine production by the Th17/Th1 cells was not investigated and their findings may be obscured by an effect of mixing non-classical Th1 with Th17/Th1 cells. Currently there are limited data regarding Th17/Th1 cells in NIU patients. An increase in CD4^+^ CD161^+^ non-classical Th1 cells or Th17/Th1 cells in the bloods from patients with juvenile idiopathic arthritis (JIA)-associated uveitis as compared with idiopathic anterior uveitis patients and healthy controls has been reported. However, no such increase was observed in JIA without uveitis [67]. This indicates a possible role for Th17/Th1 cells, and non-classical Th1 cells may solely reside within the eye and may be considered a selective target for treatment with biologics. It is known that corticosteroids suppress mainly Th1 cells and not Th17 cells [68]. On the other hand, cyclosporin A, an inhibitor of calcineurin, which is a beneficial therapeutic tool, especially in different NIUs and thrombophlebitis in course of Behçet’s disease, appears to suppress Th17 more than Th1 cells [68]. There is very little evidence available yet to determine if cyclosporin A can suppress Th17 cell plasticity although it is tempting to speculate that this will be the case. The anti-TNF-α mAb biologics (etanercept and adalimumab) have both been shown to preferentially suppress Th17/Th1 cells and Th17 cell plasticity in vivo in juvenile idiopathic arthritis [69]. In addition, Th17/Th1 cells (CD4^+^IL-17^+^CD161^+^) have been demonstrated to be sensitive to methotrexate (MTX) treatment in JIA uveitis [67]. These would explain the clinical benefits seen following anti-TNF-α and MTX therapy in JIA-associated uveitis. From this work, it is tempting to speculate that the Th17/Th1 cells contribute to the pathogenesis of retinal inflammation and act as a depot for generating effector Th1 or Th17 cells during EAU.

## 5. Importance of Treg/Th17 Cells in Autoimmune Disease and EAU

CD4^+^Foxp3^+^ regulatory T cells (Treg) possess the unique ability to protect the host from life-threatening autoimmune reactions by dampening inflammatory responses. Forkhead box protein 3 (Foxp3) has been identified as a transcription factor that characterizes Treg and is used as a marker to study its function in mice and man over the past two decades [70,71]. The majority of CD25^+^Foxp3^+^ Treg develop in the thymus under the control of TCR/CD28 stimulation and IL-2, with IL-15 signaling via a two-step model in which CD25 is induced first and then FoxP3 [72]. These Treg produce anti-inflammatory cytokines such as IL-35, TGF-β and IL-10 which suppress effector CD4^+^ T-cell responses [73]. In short, Treg modulate their regulatory function by suppressing cytolysis, modulating dendritic cell (DC) function, and by metabolic competition [74,75]. Recently, it has been reported that a newly discovered Treg subset, TIGIT^+^ Treg, promotes disease remission in autoimmune uveitis and potentially other autoimmune diseases, and this is dependent on the expression of A2Ar [66,76,77]. Treg (CD4^+^CD25^hi^ or CD4^+^FoxP3^+^) have been shown to play an important role in protecting against Vogt-Koyanagi-Harada (VKH), ocular BD, JIA-associated uveitis and other NIUs by functionally suppressing Th1 and Th17 cells [66,67,78,79]. Furthermore, in EAU models, Treg have been identified as exerting anti-inflammatory effects within the eye [35,80].

The balance of Th17 cells and Treg is thought to be important in the development of NIU and EAU [81]. In the eye, Treg and the IL-23 receptor (IL-23R) form a positive feedback loop, which serves to stabilize the pathogenic Th17 cells in EAU [82]. Another subset of Treg, CD4^+^CD25^–^CD127^low/–^ cells, co-expressing 30–50% FoxP3, were reported to be dysfunctional Tregs, in that these cells lacked proliferative potential and were unable to suppress T-cell proliferation in vitro. These cells, however, retain the characteristics of Treg with regards to their production of IL-2 [83,84]. Previous studies have demonstrated that IL-10 and TGF-β induce antigen-activated CD4^+^CD25^–^ Treg to become CD4^+^CD25^+^ Treg [85]. However, contradictory reports have indicated that in some autoimmune diseases such as systemic lupus erythematosus, these CD4^+^CD25^–^ FoxP3^+^CD127^low/–^ dysfunctional Treg are associated with disease activity [83]. On the other hand, it has been reported that these CD4^+^CD25^–^FoxP3^+^ T cells were shown to down-regulate Foxp3 and “transdifferentiate” into interleukin (IL)-17-producing effector Th17 cells under the influence of IL-6 and TGF-β [86]. Conflicting reports have suggested that CD4^+^CD25^int^Foxp3^int^RORγt^+^ cells, also characterized by CD45RA^high^, CD45RO^low^, HLADR^int^, and production of IL-10 instead of IL-17, are a subset of plastic Tregs which lose their anti-inflammatory function and do not differentiate into Th17 cells in polyposis and colon cancer patients [13]. Other reports, however, suggest that CD4^+^RORγt^+^Foxp3^+^ Treg were either in a one-way transitional process during commitment to the Treg effector lineage under high levels of TGFβ, retinoic acid and IL-2 stimulation, or to IL-17-producing Th17 cells under IL-6, IL-21, IL-23, and low levels of TGFβ [14]. These CD4^+^RORγt^+^Foxp3^+^ Treg, characterized by the expression of Sox4, CCR6, CCL20, IL-23R, and receptor activator of NF-kB ligand (RANKL, also called TNFSF11) [87], however, do not transdifferentiate into IFNγ-producing Th1 cells. In addition, the anti-inflammatory properties of those CD4^+^Foxp3^+^IL-17^+^ Treg are compromised, and they expand within a cancer environment [88]. CD4^+^Foxp3^+^IL-17^+^ Treg were reported to be pathogenic in the context of rheumatoid arthritis and IBD, inducing a local surge of IL-6 [87,89].

In the ocular context, it has been reported that circulating IL-17^+^ plastic Treg play a pathogenic role in idiopathic orbital inflammation and dry eye disease [90,91]. In a dry eye disease (DED) mouse model, it has been reported that aged Treg were prone to develop plasticity and become pathogenic, contributing to increased corneal permeability and conjunctival goblet cell loss [91]. Our previous study slightly contradicted that result, in that the levels of CD4^+^RORγt^+^Foxp3^+^ Treg in the peripheral blood did not differ between active and inactive NIU patients [66]. IL-10-producing Th17 cells were found to be increased in the resolution phase of monophasic rat EAU and decreased in number during resolution and relapse phases of the disease, indicating that they are non-pathogenic. However, freshly isolated rat effector T cells also co-expressed the markers CD25 and FoxP3, thus data showing the relationship of plastic Treg coexpressing CD25 and FoxP3 in this rat EAU model is difficult to interpret [21]. In mouse models of EAU, 10% of Treg are detectable at peak EAU, although plasticity within the Treg/Th17 axis is still unclear. The plasticity of Treg towards an effector cell phenotype may be an important factor contributing to the disease.

## 6. Concluding Remarks and Future Perspectives

It is traditionally thought that effector CD4^+^T-cells are critical for eliciting inflammation, and FoxP3^+^ Treg are important to maintain homeostasis and prevent structural damage resulting from retinal inflammation in NIU. Despite the progress in our understanding of Th1, Th17, and Treg as major players in uveitis, there remain key unanswered questions. Although these plastic CD4^+^ T cells comprised only a minor proportion during disease, their presence may, in part, explain why some individuals respond poorly to certain immunosuppressive agents. Further studies to investigate their immunopathogenicity, phenotypic plasticity and functional adaptability are necessary in order to understand their modulatory role(s) in disease.

## Figures and Tables

**Table 1 ijms-22-09584-t001:** Involvement of CD4^+^ T cells in induced EAU models in rat and mice.

Mouse Model	Method of Induction	Duration of Retinal Inflammation	Patterns of Intraocular T Cells
Monophasic Lewis rat EAU [8,31]	R16 in CFA and mTB H37Ra. PTX ip.	Effector phase: 9–18 dpi Peak phase: 14 dpi	Th17: Highest levels at peak disease, declining thereafter Treg: Levels increase at peak disease, remaining high throughout [31] Th17/Th1 and Th17/Treg: ND
Monophasic Lewis rat EAU [22]	PDSAg in CFA and mTB H37Ra.	Effector phase: 11–21 dpi Onset: 11–13 dpi Peak phase: 13–16 dpi	Th1 and Th17: levels remain stable throughout FoxP3^+^ Treg: peak at resolution Th17/Th1: Levels increase from onset, peaking at resolution phase Th17/Treg, Th1/Treg: not detected throughout
Relapsing Lewis rat EAU [22]	R14 in CFA and mTB H37Ra.	Effector phase: 7–23 dpi Onset: 7–9 dpi Peak phase: 9–13 dpi Relapse: 23 dpi	Th1: Gradually increase from onset, peaking at resolution and relapse stage Th17: Highest at onset, then gradually declining. FoxP3^+^ Treg: peak at resolution phase Th17/Th1: stable at 10% throughout Th17/Treg, Th1/Treg: not detected throughout
Monophasic and relapsing classic EAU in B10. RIII & B10.A mice [6]	IRBP_1–20_ in CFA, mTB H37Ra, +/− PTX	Onset: ~10–12 dpi Peak: 14–21 dpi Retinal atrophy phase starts 4–5 weeks pi	Th1: Dominant at peak and disease relapse [32] Th17: Dominant at induction phase Treg: Increase at peak disease and remain at high levels thereafter [33] Th17/Th1: Comprises 10% of CD4^+^ T cells at peak disease [34]
Monophasic and relapsing classic mouse EAU in C57/Bl6 [6,32]	IRBP_161–180_ in CFA, mTb H37Ra, PTX	Effective phase: 7–28 dpi Onset: ~10–12 dpi Peak: 14–21 dpi Retinal atrophy phase starts 4–5 weeks pi	Th1: Dominant at peak disease and during relapse [32]. Th17: Dominant at induction phase Treg: Appears at peak disease and increases thereafter [35] Th17/Th1: 5–10% at peak disease [36,37]. Treg/Th17 cells: ND
Chronic EAU in B10.RIII mice [24]	IRBP_1–20_ in CFA, mTb H37Ra, PTX	Effector phase: 14 dpi Peak and plateau: 10 wk pi	Peak phase: memory CD44^hi^ Th17 cells dominate with minimal Th1 cells. Treg: NDTh17/Th1: less than 1% at 12 wk pi Treg/Th17: ND
Chronic EAU in C57/Bl6 mice [24]	IRBP_1–20_ in CFA, mTb H37Ra, PTX	Effector phase: 21 dpi Peak and plateau: week 7 pi	Peak disease: memory CD44^hi^ Th17 cells dominate, with fewer than 1% Th1 cell involvement. Treg: appears at 14 dpi, gradually increasing thereafter [35] Th17/Th1: less than 1% at 12 wk pi Treg/Th17: ND
Relapsing uveitogenic DC induced-EAU [25]	Infusion of IRBP-pulsed mature DC (sc) to B10.RIII mice	Inconsistent results. Onset: 12–14 dpi. Remains active for 3 months.	Th1 and Th2 responses dominate at peak disease Th17 response is minimal Treg: ND Th17/Th1 and Th17/Treg cells: ND
IRBP-specific (R161H) B10.RIII TCR Transgenic mice [27,28]	The model generates high proportion of effector CD4^+^ T cells specific to IRBP_161–180_	Onset: 5–6 wk. Peak: 8–10 wk. Remains active for 4 months.	Th1 and Th17: both pathogenic, with Th1 cells being dominant cell type. Both polarised Th1 and Th17 cells are susceptible to EAU. Th17/Th1: presented < 1% at peak disease. Treg: Less Treg were detected intraocularly in R161H mice than the wild type EAU and they were generated extrathymically.
AIRE−/− EAU [30]	Spontaneous model of EAU due to central tolerance	Effective phase: 5–14 weeks	Th1 cells dominant and a milder Th17 response Th17/Th1 and T17/Treg cells: ND

CFA: complete Freund’s adjuvant; dpi: days post disease induction; EAU: experimental autoimmune uveitis; ND: not determined; mTB: killed mycobacterial antigen; PTX: pertussis toxin; wk: week.

## Data Availability

Not applicable.

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
