# Peer review of "CD4+ T-Cell Plasticity in Non-Infectious Retinal Inflammatory Disease"

_ijms, 2021, doi:10.3390/ijms22179584_

Round 1

Reviewer 1 Report

This is a nicely written review that discusses the current findings related to T cell plasticity in ocular inflammatory disorders. Authors nicely discuss T cell plasticity in both animal and human disease.

Table 1 summarizes the different animal models and the disease course and the involvement of different T cell types. It would be nice to graphically compare the different animal models representing in different panels the different parameters (disease score, T cell subsets etc) vs. time.

Authors suggest that the different response to treatment with corticosteroids and immunosuppressants observed in a substantial number of uveitis patients could be attributed to T cell plasticity and the different Th17/Treg Th1/Th17, etc subsets. Authors mention corticoids, MTX, but is there any evidence related to how cyclosporin modulates such response?

Author Response

We thank the reviewer for their comments. From the literature, we know that corticosteroids suppress mainly Th1 cells and not Th17 cells [1].  On the other hand, cyclosporin appears to suppress Th17 more than Th1 cells [1]. There is very little evidence available yet to determine if cyclosporin can suppress Th17 cell plasticity although it is tempting to speculate that this will be the case. Future studies are needed to develop personalized medicine in that direction. (The changes are revised between Page 6, line 229-233)

References

  1. Schewitz-Bowers LP, Lait PJP, Copland DA, Chen P, Wu W, Dhanda AD, Vistica BP, Williams EL, Liu B, Jawad S et al: Glucocorticoid-resistant Th17 cells are selectively attenuated by cyclosporine A. Proceedings of the National Academy of Sciences 2015, 112(13):4080-4085.

Reviewer 2 Report

In the present manuscript entitled "Function of CD4+ T-cell plasticity in non-infectious retinal inflammatory disease", Chen and colleagues offer a review considering the dichotomy of labor of different CD4+ T-cell subsets in the autoimmune uveitis.

The manuscript is well written, discussing relevant references and pointing the relevance for further investigations about the immunopathogenicity, phenotypic plasticity and functional adaptability of these subsets to better understand their potential regulation avoiding the use of traditional  immunosuppressive therapies

Author Response

Thank you to your comments!